# Gender effects on outcomes of psychosomatic rehabilitation are reduced

**Juliane Burghardt**[1]*, **Manuel Sprung**[1,2,3]

**1** Division of Clinical Psychology, Department Psychology and Psychodynamics, Karl Landsteiner University of Health Science, Krems an der Donau, Austria, **2** University Hospital for Psychosomatic Medicine Eggenburg, Psychosomatisches Zentrum Waldviertel, Eggenburg, Austria, **3** Psychiatric Rehabilitation Clinic Gars am Kamp, Psychosomatisches Zentrum Waldviertel, Gars am Kamp, Austria

* Juliane.Burghardt@kl.ac.at

## Abstract

### Objective

The study examined whether psychiatric/psychosomatic rehabilitation continues to have a better course of treatment for women than men.

### Methods

We compared the course of global symptom severity, health-related quality of life and functioning between admission and discharge in patients (848 men, 1412 women) at an Austrian psychiatric/psychosomatic rehabilitation clinic.

### Results

Gender-specific differences in the course of treatment were all too small to be clinically relevant. The differences were smallest in the middle-aged cohort. However, at the time of admission, women reported a slightly higher symptom burden.

### Conclusion

Overall, the results show a gender-fair effectiveness of the rehabilitation. The new findings could be explained by changes in living conditions, gender roles, or better treatment methods.

## Introduction

Psychosomatic rehabilitation focusses on the psychotherapeutic treatment of mental disorders in a stationary context with the goal to reduce symptoms and increase functioning and quality of life. Many studies have shown that psychosomatic rehabilitation shows better treatment effects for women than for men [1–3]. This treatment advantage for women is not limited to psychosomatic rehabilitation interventions, but is reflected in a number of rehabilitation settings (see [4]) e.g. for chronic back pain [5] or the treatment of cardiovascular diseases [6]. The meta-analytical MESTA study confirmed the better treatment effects for women in

**Data availability statement:** All relevant data are within the manuscript and its Supporting information files.

**Funding:** The Lower Austria research and education company (NFB) (Niederösterreichische Forschungs- und Bildungsges.m.b.H.; NFB) supported this research by financing the endowed professorship in clinical psychology held by MS. The Open Access Publishing Fund of Karl Landsteiner University of Health Sciences, Krems, Austria supports the publication of this manuscript. The VAMED Institute for Gender Medicine financed the position held by JB. The funders had no role in study design, data collection and analysis, decision to publish, or preparation of the manuscript.

**Competing interests:** Prof. Dr. Manuel Sprung is the scientific director of the University Clinic for Psychosomatic Medicine Eggenburg, PSZW (Eggenburg and Gars). This does not alter our adherence to PLOS ONE policies on sharing data and materials.

psychosomatic rehabilitation interventions [3]: An increase in the proportion of women in the patient sample was accompanied by an improvement in the treatment outcome between admission and discharge (β = .22). Later studies also replicated this effect of psychosomatic rehabilitation. For example de Vries and colleagues [1] showed a greater reduction in symptoms (depression, psychological stress, and psychosocial health) for female patients than for male patients with data from 2008 to 2010. A more recent study with data from 2013 could not find any significant differences in the effectiveness of rehabilitation in terms of symptom burden or resilience [7]. Only self-regulatory ability and work motivation showed the effect of gender on treatment outcomes. Multiple reasons for the gender differences were suggested. It was argued that that rehabilitation interventions are better tailored to the needs of women than to the needs of men [2]. These different needs are based on gender roles [8,9]. Among other things, seeking help and support contradicts the male gender role, which is why men often show less help seeking behavior than women [10] and communicate less effectively with health care providers. For instance, some men may understate pain or hide emotions [8] or are less accurate about their medical records [11]. The loss of autonomy during treatment is possibly more problematic for men than for women [12]. The patient role is incongruent with the male gender role [13], which is active and agentic. In line with the male gender role, men prefer a higher level of control over the therapeutic process and report a need for action-oriented problem-solving strategies [12]. Accordingly, therapy and rehabilitation should be implemented in a gender-sensitive manner [12,14]. Notwithstanding these recommendations, a study of psychiatric/psychotherapeutic treatments in 2007 concluded that interventions are largely gender-neutral [15].

Gender differences are not only evident in the course of treatment; women and men differ in the severity of symptoms [16], as well as in the frequency [9,17], and in the course of mental disorders [18]. The greater treatment effects in women could therefore be explained by the greater burden of symptoms upon admission [4]. In addition, the living conditions and thus the resources of men and women also differ, e.g. characterized by lower employment of women and a higher burden of childcare [19,20]. Gender roles have been changing significantly for years. As a result, women increasingly describe themselves as more masculine and less feminine than before [21,22]. The living conditions of men and women have also become more and more equal in the last few decades due to the increase in the employment of women and rising incomes [23]. As a consequence of the changed gender roles and the change in living conditions, the question must be asked whether the gender differences can still be replicated. Do women continue to benefit more from psychosomatic rehabilitation programs than men? To answer this question, the present study analyzes gender-specific differences in the effects of inpatient psychosomatic/psychiatric rehabilitation in an Austrian rehabilitation clinic. Since older cohorts often have more traditional living conditions with larger gender differences [9], an additional analysis compares gender differences in different age groups. If the gender differences decrease due to social changes, younger cohorts could show smaller gender differences than older cohorts.

## Materials and methods

### Participants

The present study is a retrospective analysis of data collected as part of the routine examination. The study analyses data from 2260 rehabilitation patients, with complete examination results (i.e. all primary outcomes in the admission and discharge surveys) who were treated between July 2011 and January 2015. Depending on the outcome, measurements from 71 to 66% of the total patient sample are available. Missing measurements are mainly due to the

**Table 1. Sample characteristics by age cohort and sex.**

| | Sex | | WHO age cohorts | | |
|---|---|---|---|---|---|
| | **male** | **female** | **<35** | **35–55** | **>55** |
| **Frequency** | 848 | 1412 | 270 | 1628 | 362 |
| **Age** M (SD) | 46.7 (8.9) | 45.8 (8.8) | 28.8 (4.0) | 46.5 (5.18) | 57.3 (2.5) |
| **Sex** | 37.5% | 62.5% | - | - | - |
| male | - | - | 10.7% | 70.6% | 18.6% |
| female | - | - | 12.7% | 72.9% | 14.4% |
| **Diagnosis*** | | | | | |
| F01-09 | 4 (0.5%) | 1 (0.1%) | 0 (0.0%) | 1 (0.1%) | 4 (1.1%) |
| F10-19 | 8 (1.0%) | 11 (0.8%) | 5 (1.9%) | 13 (0.8%) | 1 (0.3%) |
| F20-29 | 16 (1.9%) | 21 (1.5%) | 10 (3.7%) | 27 (1.7%) | 0 (0.0%) |
| F30-39 | 528 (63.2%) | 892 (63.9%) | 140 (52.4%) | 1035 (64.5%) | 245 (68.1%) |
| F40-48 | 228 (27.3%) | 403 (28.9%) | 91 (34.1%) | 447 (27.9%) | 93 (25.8%) |
| F50-59 | 4 (0.5%) | 6 (0.4%) | 1 (0.4%) | 4 (0.2%) | 5 (1.4%) |
| F60-69 | 12 (1.4%) | 23 (1.6%) | 15 (5.6%) | 19 (1.2%) | 1 (0.3%) |
| F70-79 | 0 (0.0%) | 0 (0.0%) | 0 (0.0%) | 0 (0.0%) | 0 (0.0%) |
| F80-89 | 0 (0.0%) | 0 (0.0%) | 0 (0.0%) | 0 (0.0%) | 0 (0.0%) |
| F90-98 | 2 (0.2%) | 0 (0.0%) | 2 (0.7%) | 0 (0.0%) | 0 (0.0%) |
| E00-90 | 3 (0.4%) | 0 (0.0%) | 0 (0.0%) | 2 (0.1%) | 1 (0.3%) |
| I00-99 | 1 (0.1%) | 0 (0.0%) | 0 (0.0%) | 1 (0.1%) | 0 (0.0%) |
| M00-99 | 0 (0.0%) | 1 (0.1%) | 0 (0.0%) | 1 (0.1%) | 0 (0.0%) |
| Z73 | 29 (3.5%) | 38 (2.8%) | 3 (1.1%) | 54 (3.4%) | 10 (2.8%) |
| **Employment**+ | | | | | |
| unemployed | 190 (28.9%) | 294 (28.1%) | 75 (36.2%) | 343 (28.1%) | 66 (23.9%) |
| retired# | 33 (5.0%) | 69 (6.6%) | 12 (5.8%) | 63 (5.2%) | 27 (9.8%) |
| employed | 434 (66.1%) | 682 (65.3%) | 120 (58.0%) | 813 (66.7%) | 183 (66.3%) |

* **F01-F09** Mental disorders due to known physiological conditions, **F10-F19** Mental and behavioral disorders due to psychoactive substance use, **F20-F29** Schizophrenia, schizotypal, delusional, and other non-mood psychotic disorders, **F30-F39** Mood disorders, **F40-F48** Anxiety, dissociative, stress-related, somatoform and other nonpsychotic mental disorders, **F50-F59** Behavioral syndromes associated with physiological disturbances and physical factors, **F60-F69** Disorders of adult personality and behavior, **F70-F79** Intellectual disabilities, **F80-F89** Pervasive and specific developmental disorders, **F90-F98** Behavioral and emotional disorders with onset usually occurring in childhood and adolescence, **F99-F99** Unspecified mental disorder; E00-E90 Endocrine, nutritional and metabolic diseases; **I00-I99** Diseases of the Circulatory System; **M00-M99** diseases of the musculoskeletal system and connective tissue; **Z73** burnout (the information on the main diagnoses of 29 patients is missing in the database).

+ Information from 29 patients (1.3% of the sample) was missing.

# Pension, has applied for pension payment, rehabilitation pay or sick pay.

clinic's internal processes. The sample comprised 848 (37.5%) men and 1412 (62.5%) women. The patients were between 18 and 74 years old at the time of admission. Their mean age was 46.1 years (SD = 8.8; median = 47.7). Details are given in Table 1. Male patients were on average somewhat older than female patients, $t(2258) = 2.251$; $p < 0.05$, 95% CI [0.11; 1.61], Cohen's $d_s = 0.10$. The most common diagnoses (main diagnosis) of patients were F30-F39 mood disorders (63.6%) and F40-F48 neurotic, stress and somatoform disorders (28.3%). At the time of admission, 1,116 (65.6%) of the patients were employed, 484 (28.4%) were unemployed and 102 (6.0%) were retired or had applied for a pension or were receiving rehabilitation or sick pay. Gender and age-specific differences with regard to diagnoses and occupational status (employment) are described in Table 1. Male and female patients did not differ significantly with regard to diagnosis or occupational status. Further information on the total sample can be found in Riffer and colleagues [24–26].

## Materials

**Basic socio-demographic and clinical data.** Gender, age, diagnoses (i.e. the main diagnoses upon discharge) and information on the professional status (employment) were determined using the basic documentation of the hospital information system. Gender (male, female) was defined in terms of biological sex. Age referred to the chronological age (in years) at the time of admission. Diagnoses were classified using the ICD-10 [27].

**Symptom burden, quality of life and functional ability.** The routine examination survey assessed the general symptom burden, quality of life and functional ability (primary outcomes). General symptom severity was determined using the Symptom Checklist (SCL-90) [28] or the Brief Symptom Checklist (BSCL) [29]. The checklists measure the subjectively perceived impairments caused by physical and psychological symptoms in a total of nine symptom areas with 90, respectively 53 items on a 5-point response scale. The BSCL had been derived from the SCL-90 by selecting the items with the highest item loading. Previous data showed that the BSCL and SCL-90 are highly correlated [r = .92 to.99], [30]. Both instruments provide a Global Severity Index (GSI), which is a commonly used measure of general psychological distress. Items were averaged to form the index. Higher values imply higher symptom severity. The quality of life was determined with the WHO Quality of Life questionnaire (WHOQOL-BREF) [31]. It assesses the subjective health-related quality of life in four sub-areas as well as globally using 26 items (with a 5-point response scale). The quality of life was determined by adding values for each area. This provided measures for physical, psychological, social, and environmental quality of life as well as a global value. Higher scores denote better quality of life. Functional ability was determined using the Global Assessment of Functioning (GAF) scale of the Diagnostic and Statistical Manual of Mental Disorders (DSM-IV) [32]. Based on external assessments by the attending physician, the GAF is used to determine a single value for the current general functional level of the patient (graded into 10 levels). Higher scores imply better functioning. Relevant comparative values of other Austrian psychiatric rehabilitation patients can be found in a current meta-analysis [33].

## Procedure

The study used a naturalistic one-group pre-post design without a control group. The analysis used data of patients who were treated at the psychiatric rehabilitation clinic Gars am Kamp in Austria. Patients answered the questionnaires in a computer assessment room in the presence of a trained professional. The self-report measures were assessed using the Hogrefe test system. This system provides a platform that administers standardized questionnaires licensed by Hogrefe. It provides a user-friendly surface and assures data integrity. The GAF was assessed during individual medical examinations.

The analyses were approved by the ethics commission of the Karl Landsteiner University of Health Sciences (Nr: 1006/2021). We complied with the requirements of the current version of the Declaration of Helsinki [34]. The patients agreed to the data collection and usage when they started treatment, all analyses were conducted on pseudonymized data.

## Treatment

All patients in the study sample took part in a standard multidisciplinary therapy program of 22½ hours per week during a planned 6-week inpatient stay at a psychiatric/psychosomatic rehabilitation clinic in Austria. The therapy, which largely takes place in open groups that are not specific to the disorder, was carried out by a multidisciplinary treatment team in accordance with the treatment plan of the pension insurance institution (Pensionsversicherungsanstalt; www.pv.at) responsible for psychiatric/psychosomatic rehabilitation in Austria.

Psychosomatic rehabilitation in Germany and psychiatric rehabilitation in Austria are very similar approaches to treat mental disorders. Both emphasize psychotherapeutic interventions, but also include psychopharmacological and various other complementary interventions, for instance excise and physical therapy. Both mainly treat patients with depressive or anxiety-related disorders. Further details on the treatment program can be found elsewhere [25]. Due to the legal requirements applied to all psychiatric rehabilitation clinics in Austria the treatment program is comparable to that of other Austrian rehabilitation clinics [33]. The length of stay of the patients in the study sample was between 39 and 62 days (M = 42.1; SD = 3.5; Modus = 41.0). There were no significant gender- or age-specific differences in the length of stay.

## Analyses

The data was extracted from the Hogrefe test system and the clinic information system. To simplify analysis, only complete data sets were evaluated. The analyzes tested for gender-specific differences at the beginning of treatment (admission survey) and for gender-specific differences in the changes in the examination results (primary outcomes) between admission and discharge. The final sample had a power of over .99 to find small effects for this interaction between gender and measurement time (G*Power Version 3.1.9.2., [35]). The gender-specific pre-post and baseline values at admission were compared with t-tests and repeated-measure ANOVAs using SPSS 26. These treatment effects were tested both in the overall sample and separately for three age cohorts. The level of significance was set at α <0.05 (two-sided). Cohen's d values above 0.8 are interpreted as large effects, between 0.5 and 0.2 as medium effects and effects below 0.2 are interpreted as small effects [36]. Corresponding values for $\eta_p^2$ recommend that values from 0.14 should be interpreted as large, from 0.06 as medium and from 0.01 as small effects [37].

# Results

## Results at pre- (T1) and posttest (T2), separately for each gender

To test the effectiveness of the treatment, we compared the outcomes of T1 (pretest) and T2 (posttest) separately by gender. The analyzes in Table 2 show that all outcomes at T2 improved compared to T1, for both men and women. The changes show small to large effects. The biggest effects were found for functional ability (GAF), the smallest in the areas of social and environmental quality of life (QOL$_{social}$ and QOL$_{environment}$).

## Outcome comparisons at admission (T1)

The level of distress at the time of admission (T1) was then compared between men and women. Table 3 contains the corresponding t-tests for independent samples at admission (T1). In line with earlier findings, women were more distressed than men at admission. This difference was most pronounced in the psychological area of quality of life (QOL$_{mental}$) and general symptom burden (GSI). Significant differences were also found in the areas of environment (QOL$_{environment}$), physical (QOL$_{physical}$), and global quality of life (QOL$_{global}$).

## Gender differences in changes between T1—T2

The course of treatment for men and women was compared with the help of a repeated measures ANOVAs, controlling for the differences in symptom burden at the time of admission (T1). Table 4 reports the results of the 2 (time) × 2 (gender) ANOVAs per outcome variable. Treatment effects were gender-specific (= significant interaction between the factors time × gender) for general symptom burden (GSI) and quality of life in the physical and psychological domain (QOL$_{physical}$, QOL$_{mental}$), as well as functional ability (GAF). Functional ability

**Table 2. Treatment outcomes for men and women.**

| Outcomes | T1 | | T2 | | t-test | | | Corr. | Effect size |
|---|---|---|---|---|---|---|---|---|---|
| | *M* | *SD* | *M* | *SD* | *t* | *df* | *p* | *r* | *dav* |
| [↓]GSI | | | | | | | | | |
| men | 1.10 | 0.67 | 0.80 | 0.68 | 13.78 | 847 | < 0.001 | 0.55 | 0.44 |
| women | 1.31 | 0.66 | 0.87 | 0.70 | 24.0 | 1411 | < 0.001 | 0.52 | 0.65 |
| [↑]QOL$_{physical}$ | | | | | | | | | |
| men | 55.52 | 18.31 | 64.40 | 20.95 | -17.073 | 847 | < 0.001 | 0.71 | 0.45 |
| women | 51.80 | 18.77 | 62.62 | 19.91 | -26.51 | 1411 | < 0.001 | 0.69 | 0.56 |
| [↑]QOL$_{psychological}$ | | | | | | | | | |
| men | 48.91 | 19.18 | 58.48 | 21.01 | -17.589 | 847 | < 0.001 | 0.69 | 0.48 |
| women | 41.18 | 18.83 | 53.36 | 20.32 | -28.125 | 1411 | < 0.001 | 0.66 | 0.63 |
| [↑]QOL$_{social}$ | | | | | | | | | |
| men | 55.33 | 22.83 | 61.32 | 23.09 | -9.55 | 847 | < 0.001 | 0.68 | 0.26 |
| women | 55.99 | 23.12 | 61.77 | 22.79 | -11.14 | 1411 | < 0.001 | 0.64 | 0.25 |
| [↑]QOL$_{environment}$ | | | | | | | | | |
| men | 69.44 | 15.93 | 71.32 | 17.14 | -4.54 | 847 | < 0.001 | 0.74 | 0.11 |
| women | 65.42 | 16.15 | 67.75 | 16.72 | -7.10 | 1411 | < 0.001 | 0.72 | 0.14 |
| [↑]QOL$_{global}$ | | | | | | | | | |
| men | 45.71 | 21.11 | 58.13 | 22.66 | -17.68 | 847 | < 0.001 | 0.57 | 0.57 |
| women | 42.69 | 21.46 | 56.70 | 22.31 | -24.48 | 1411 | < 0.001 | 0.52 | 0.64 |
| [↑]GAF | | | | | | | | | |
| men | 59.68 | 7.12 | 66.51 | 8.46 | -38.461 | 847 | < 0.001 | 0.79 | 0.88 |
| women | 59.45 | 6.67 | 65.81 | 7.82 | -45.72 | 1411 | < 0.001 | 0.75 | 0.88 |

[*] lower value = positive;

[*] higher value = positive.

Corr = correlation between T1 and T2.

showed a more pronounced improvement in men; the other two measures showed greater effects in women. At the same time, all effect sizes were so small that the differences should be viewed as clinically insignificant.

## Gender-specific differences in the change between T1—T2 per age cohort

For further exploration, we tested whether the gender effects differed between age cohorts. As the living conditions of men and women become increasingly similar, it would be possible that older cohorts show greater gender differences than younger ones. To test this, we repeated the repeated

**Table 3. Group comparison of outcomes at admission (time T1).**

| Outcomes | sex (male vs. female) | | | |
|---|---|---|---|---|
| | *t* | *df* | *p* | *ds* |
| **GSI** | -7.059 | 2258 | < .001 | 0.32 |
| **QOL$_{physical}$** | 4.605 | 2258 | < .001 | 0.20 |
| **QOL$_{psychological}$** | 9.377 | 2258 | < .001 | 0.41 |
| **QOL$_{social}$** | -0.665 | 2258 | .506 | < 0.001 |
| **QOL$_{environment}$** | 5.748 | 2258 | < .001 | 0.25 |
| **QOL$_{global}$** | 3.260 | 2258 | < .01 | 0.14 |
| **GAF** | 0.781 | 2258 | .435 | < 0.005 |

**Table 4. Group comparison in the improvement of the outcomes of admission and discharge surveys (T1—T2).**

| Outcomes | 2 (time) × 2 (sex) ANOVA | | | | | 2 (time) × 2 (sex) ANOVA by age cohort | | | | |
|---|---|---|---|---|---|---|---|---|---|---|
| | Factor | $F$ | $df$ | $p$ | $\eta^2$ | Age cohorts | $F^{+}$ | $df$ | $p$ | $\eta^2$ |
| **GSI** | time | 650.28 | 1 | < .001 | 0.22 | <35 | 4.17 | 1 | .042 | 0.015 |
| | sex | 30.44 | 1 | < .001 | 0.013 | 35–55 | 9.34 | 1 | .002 | 0.006 |
| | time × sex | 20.76 | 1 | < .001 | 0.009 | >55 | 10.63 | 1 | .001 | 0.029 |
| **QOL**<sub>physical</sub> | time | 882.19 | 1 | < .001 | 0.28 | <35 | 1.82 | 1 | .179 | 0.007 |
| | sex | 12.48 | 1 | < .001 | 0.005 | 35–55 | 2.88 | 1 | .090 | 0.002 |
| | time × sex | 8.59 | 1 | .003 | 0.004 | >55 | 6.22 | 1 | .013 | 0.017 |
| **QOL**<sub>psychological</sub> | time | 965.40 | 1 | < .001 | 0.30 | <35 | 10.70 | 1 | .001 | 0.038 |
| | sex | 66.93 | 1 | < .001 | 0.03 | 35–55 | 2.02 | 1 | .156 | 0.001 |
| | time × sex | 13.81 | 1 | < .001 | 0.006 | >55 | 10.86 | 1 | .001 | 0.029 |
| **QOL**<sub>social</sub> | time | 202.51 | 1 | < .001 | 0.08 | <35 | 3.48 | 1 | .063 | 0.013 |
| | sex | 0.37 | 1 | n.s. | - | 35–55 | 1.50 | 1 | .221 | 0.001 |
| | time × sex | 0.07 | 1 | n.s. | - | >55 | 0.36 | 1 | .548 | 0.001 |
| **QOL**<sub>envir.</sub> | time | 62.82 | 1 | < .001 | 0.03 | <35 | 7.65 | 1 | .006 | 0.028 |
| | sex | 32.47 | 1 | < .001 | 0.01 | 35–55 | 0.18 | 1 | .675 | < 0.001 |
| | time × sex | 0.72 | 1 | n.s. | - | >55 | 0.69 | 1 | .407 | 0.002 |
| **QOL**<sub>global</sub> | time | 829.75 | 1 | < .001 | 0.27 | <35 | 3.64 | 1 | .058 | 0.013 |
| | sex | 7.14 | 1 | .008 | 0.003 | 35–55 | < 0.01 | 1 | .959 | < 0.001 |
| | time × sex | 2.99 | 1 | n.s. | - | >55 | 8.26 | 1 | .004 | 0.022 |
| **GAF** | time | 3400.07 | 1 | < .001 | 0.60 | <35 | 5.16 | 1 | .024 | 0.019 |
| | sex | 2.34 | 1 | n.s. | - | 35–55 | 4.13 | 1 | .042 | 0.003 |
| | time × sex | 4.25 | 1 | .039 | 0.002 | >55 | 1.04 | 1 | .309 | 0.003 |

+F-values for interaction time by sex.

measures ANOVAs for the three age groups. The age groups were formed in accordance with the WHO age limits and corresponded to the ranges 18–35 years = young adults, 35–55 years = middle-aged adults, over 55 years = older adults. If the low gender effects were based on cohort effects, the oldest cohort should show the largest gender-specific treatment effects. The results did not match these expectations. Gender differences in treatment effects were slightly larger in both the youngest and oldest age groups than for the entire sample. The results are shown in Table 4. The youngest age group showed gender differences in the course of treatment, with at least a small effect size ($\eta^2 > 0.01$) in the general symptom burden, the quality of life in the psychological, social and environmental domains as well as in the global evaluation (QOL<sub>mental</sub>, QOL<sub>social</sub>, QOL<sub>environment</sub>, QOL<sub>global</sub>). These measurements showed a better course of treatment in women. Functional ability (GAF) also showed a small gender difference, however, men showed the better course in this measure. The oldest cohort also reported a more positive course of treatment for women of at least small size in terms of general symptom burden and quality of life in the physical, mental and global domains (QOL<sub>physical</sub>, QOL<sub>mental</sub>, QOL<sub>global</sub>). The middle age group showed no (clinically relevant) gender differences over the course of treatment.

## Discussion

Both men and women showed substantial improvements in all examined outcome areas (general symptom burden, quality of life and functional ability). Although some of these improvements showed significant gender-specific differences, the effect sizes of these differences were so small that they can be regarded as clinically negligible. Older findings that showed moderate gender differences in the course of treatment could therefore not be replicated. Substantial

gender differences were neither evident in externally rated functional ability (GAF) nor in self-report measures. At the time of admission, women showed greater symptom burden than men. The gender differences in treatment outcomes were most pronounced in the oldest and youngest age group, but remained small in these groups too. Accordingly, all age groups, both men and women, showed broad treatment success. The age group patterns did not match a linear decrease in the gender difference in the sense of a cohort effect. If the equalization of living conditions between men and women were the reason for the decrease in gender differences, it should be least pronounced in the youngest age group. The interpretation is complicated, however, by the fact that the analysis confounded age and cohort effects. The data pattern fits an explanation of gender differences through gender roles. The relatively low gender effect in the middle age group could be based on the gender differences being driven by masculine gender role orientations. These show the smallest differences between men and women in middle adulthood [38,39]. However, to test this explanation a direct measure of gender role orientation would be necessary. Since most patients in this study fall within the middle age group the very small overall gender effect is driven by this age group. However, previous meta-analyses of psychosomatic rehabilitation interventions show that this age distribution is representative of psychiatric rehabilitation clinics in Austria and psychosomatic rehabilitation clinics Germany [3,33].

Earlier studies faced the question of whether women only benefited more from rehabilitation interventions because they were more distressed than men when they were admitted [4]. In lieu of the current data, this interpretation appears to be less likely, since there are no longer any gender differences in treatment outcomes, but women are still more distressed at admission. This suggests that the earlier gender difference in treatment outcomes was more likely due to an earlier lack of fit between needs and services; as previously suggested [2,4]. The most positive interpretation of the results would be that treatment methods are now more optimally aimed at both men and women. However, this interpretation is less in line with the larger gender differences in the youngest and oldest cohort. Alternatively, the gender differences that influenced the course of therapy differently, such as differences in gender roles or living conditions, could have decreased sufficiently in the population in certain age groups. The results may be surprising in view of the amount of evidence that has highlighted the extent of the differences in disease rates between men and women. Gender-specific differences in psychopathology are favored by a number of biological, cultural, cognitive, and affective factors and manifest themselves in a variety of ways [40]. For example, depression in men is more often characterized by a mixture of internalization (anxiety, depression) and externalization disorders (substance abuse, aggression) [41] while women report more somatic symptoms [42]. The gender-specific differences in the symptoms of depression have been neurobiologically [43] and neuroendocrinologically confirmed [44]. The depression pattern typical for men was more strongly characterized by alcohol abuse and suicide than that of women. These gender differences could be missing in the present study, since patients represent a selective sample. Previous work has noted that the gender differences could be partly attributed to biases within measures [18]. Since men and women often report different symptoms questionnaires can be biased towards capturing traditionally 'feminine' symptoms of a disorder. Therefore, it is notable that our study used the SCL-90 lists, which is the most common instrument used in studies within the MESTA-meta-analysis, which did find a gender effect $\beta$ = .22 [3]. It is therefore unlikely that our results hinge on measurement biases.

Even though, the absence of pronounced gender effects on efficacy may seem surprising, the present results only superficially contradict more recent findings. Although earlier studies continued to confirm the existence of gender differences in psychosomatic rehabilitation interventions, the corresponding effects were small for data from 2008 to 2010 [1] or limited

to a few variables for data from 2013 [7]. The greatest effect was found in work motivation, although this effect also remained small. Work motivation and related procedures could show greater effects than, for example, symptom burden, because there are gender-typical differences in working environments. Accordingly, it would make sense to investigate further treatment outcomes in future studies. It should also be examined whether other rehabilitation measures continue to show gender effects.

The strengths of the present study are the sample size and the inclusion of heterogeneous patient groups. An important extension of the present study would be the examination of post-treatment follow-up data. Previous findings showed that gender differences at follow-up assessment compared to the discharge assessment disappeared [3]. Although men benefit more from rehabilitation at discharge assessment, men may show poorer results at the follow-up assessment. Since men show more health risk behavior such as substance abuse or sleep deprivation [45], therapeutic effects could decrease especially strongly after rehabilitation. One of the limitations of the present study is that gender role orientation was not studied. The division into age groups was done post hoc and confounded age and cohort effects. Additionally, the data was collected between 2011 and 2015. It is thus, plausible that new changes have occurred. This and the fact that the data was only gathered in one clinic makes it necessary to replicate our findings. Another limitation is that no control group was included in the study. Therefore, it cannot be ruled out that without treatment the effects would be lower and that the observed gender-specific differences in the change in T1—T2 could also be found in a control group without treatment. However, a controlled study on psychiatric/psychosomatic rehabilitation in Austria confirmed that the improvements in psychiatric/psychosomatic rehabilitation are systematically greater than in an untreated control group [46]. Unfortunately, gender differences were not investigated in this study. In the future, more controlled studies should be carried out that also take gender-specific differences into account.

In contrast to earlier studies, gender-specific differences in the outcomes of psychiatric/psychosomatic rehabilitation were scarcely detectable. The present findings confirm that rehabilitation interventions can be equally effective among men and women, even though women remain more distressed upon admission than men. Thus, the higher symptom burden of women at admission does not necessarily lead to a difference in treatment effectiveness. It remains unclear whether the decline in gender differences on treatment effectiveness was caused by changes in society as a whole or by specific characteristics of the present treatment. The extent of the gender differences in rehabilitation outcomes was particularly small in the middle age group, which is consistent with the typical course of changes in gender roles over the lifespan. This would correspond to an explanation by changes in society as a whole. If the finding of comparatively bigger gender effects among older or younger patients replicates, this might imply a need for gender and age-sensitive treatments instead of a gender-sensitive treatment that ignores age.

## Supporting information

**S1 File. Evaluation data.**
(SAV)

## Acknowledgments

The results of the present study were in part presented during a poster presentation at the 18th annual conference of the Austrian Society for Psychiatry, Psychotherapy and Psychosomatics.

## Author contributions

**Conceptualization:** Juliane Burghardt, Manuel Sprung.

**Data curation:** Manuel Sprung.

**Formal analysis:** Juliane Burghardt, Manuel Sprung.

**Funding acquisition:** Manuel Sprung.

**Methodology:** Manuel Sprung.

**Project administration:** Manuel Sprung.

**Resources:** Manuel Sprung.

**Supervision:** Manuel Sprung.

**Validation:** Juliane Burghardt.

**Writing – original draft:** Manuel Sprung.

**Writing – review & editing:** Juliane Burghardt, Manuel Sprung.

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
