## [Decision Letter · Decision Letter 0]

17 Jun 2021

PONE-D-20-37367

Gender effects on outcomes of psychosomatic rehabilitation are reduced

PLOS ONE

Dear Dr. Burghardt,

Thank you for submitting your manuscript to PLOS ONE. After careful consideration, we feel that it has merit but does not fully meet PLOS ONE’s publication criteria as it currently stands. Therefore, we invite you to submit a revised version of the manuscript that addresses the points raised during the review process.

If applicable, we recommend that you deposit your laboratory protocols in protocols.io to enhance the reproducibility of your results. Protocols.io assigns your protocol its own identifier (DOI) so that it can be cited independently in the future. For instructions see: http://journals.plos.org/plosone/s/submission-guidelines#loc-laboratory-protocols . Additionally, PLOS ONE offers an option for publishing peer-reviewed Lab Protocol articles, which describe protocols hosted on protocols.io. Read more information on sharing protocols at https://plos.org/protocols?utm_medium=editorial-email&utm_source=authorletters&utm_campaign=protocols.

We look forward to receiving your revised manuscript.

Kind regards,

Stephan Doering, M.D.

Academic Editor

PLOS ONE

Journal Requirements:

https://journals.plos.org/plosone/s/file?id=wjVg/PLOSOne_formatting_sample_main_body.pdfand

"Prof. Dr. Manuel Sprung is the scientific director of the University Clinic for Psychosomatic Medicine Eggenburg, PSZW (Eggenburg and Gars). Primarius Dr. Riffer is the medical director of the PSZW."

Please confirm that this does not alter your adherence to all PLOS ONE policies on sharing data and materials, by including the following statement: "This does not alter our adherence to PLOS ONE policies on sharing data and materials.” (as detailed online in our guide for authors http://journals.plos.org/plosone/s/competing-interests ). If there are restrictions on sharing of data and/or materials, please state these. Please note that we cannot proceed with consideration of your article until this information has been declared.

Reviewers' comments:

Reviewer's Responses to Questions

**Comments to the Author**

1. Is the manuscript technically sound, and do the data support the conclusions?

Reviewer #1: Yes

Reviewer #2: Partly

2. Has the statistical analysis been performed appropriately and rigorously?

Reviewer #1: Yes

Reviewer #2: Yes

3. Have the authors made all data underlying the findings in their manuscript fully available?

Reviewer #1: Yes

Reviewer #2: Yes

4. Is the manuscript presented in an intelligible fashion and written in standard English?

Reviewer #1: Yes

Reviewer #2: Yes

5. Review Comments to the Author

Reviewer #1: Summary and overall impression

The present study investigated whether gender differences in the effectiveness of treatment of mental illness are evident in psychosomatic rehabilitation. Previous studies showed such an effect, although the evidence for this has been less clear in studies conducted in recent years. The present work shows that gender differences are partly present but clinically negligible. Differences are evident in very young and very old patients, although also small. The authors discuss different reasons why gender differences are less apparent compared with the past. The paper thus represents an important work in the field of psychosomatic rehabilitation, as it demonstrates that treatment is almost as effective for women as for men compared to the past. The sample size in all three age cohorts and the inclusion of measures of self-assessment and external assessment in the analysis are strengths of the study. Nonetheless, the fact that there are many more individuals in the middle age cohort would be important to address more in the discussion. This uneven distribution of age cohorts has a large impact on the overall lack of differences in the sample. As the authors themselves describe, it would also have been desirable if statements about gender role orientation had been possible, as these appear to be highly relevant to the interpretation of the results. Overall, the paper is very clear and well structured. It provides interpretative information on many relevant points and represents an important gain in knowledge concerning the effectiveness of psychosomatic rehabilitation.

Specific improvements

Major issues

- The discussion should strongly consider that the middle age group contained significantly more individuals, which reduced the gender effects so much in the overall analysis, although there are at least small differences in the other two age groups. I would like to see a more nuanced discussion of this.

Minor issues

- Page 3 “Among other things, seeking help […]“: n my view, the sentence does not fit well as an explanation for the effects of gender role in rehabilitation, since the individuals have already sought help and started rehabilitation. Or are there findings that men then take the treatment itself less seriously or make less use of it during inpatient treatment?

- Page 4: “Women suffer more often […]“: This sentence seems in the wrong place, since it seems that it does not contribute to the linked arguments in the sentences before and after this sentence.

- Page 6: I would recommend to put the G*Power analysis together with a more detailed description of the conducted analyses in the Analyses section. Especially the repeated-measure ANOVAs could be described more in detail.

- Page 9: “The biggest effects were found in for functional ability […]”

- Page 10: “[…] men showed the better course in this measure, which might be because […]”: Since the second half of this sentence is an interpretation, it should be moved to the discussion.

- Page 24: Table 2 needs some revision in the section “Employment”. The percentage of employed people in the oldest WHO group is in the wrong line and is missing it’s percent sign.

- The authors should point out in the limitations, that the data was obtained only from one clinic and therefore generalization is limited. Furthermore, they should discuss that the data of some patients is ten years old and gathered over a long period of time, where e.g. changes in the understanding of gender roles in the society could have changed slightly and therefore confounded the analyses.

Reviewer #2: This is an interesting paper about the effects of gender on the treatment outcomes of inpatient psychiatric rehabilitation in Austria. The paper is fairly clear in the approach and analysis. The manuscript would benefit from a few changes for readability and clarity.

Introduction

*Please describe psychosomatic rehabilitation- how is this the same or different from psychiatric rehabilitation? It may be helpful to use either one term or the other- the term “psychiatric” is preferred to describe hospital treatment.

* Please use rehabilitation interventions not rehabilitations

p.4 Consider mentioning the subjective nature diagnosis – are there studies that examine the effect of clinician gender on treatment outcomes?

Method

*There is some duplication in description of study design.

*Please address why the BriefSCL and SCLwere both utilized and the comparability of results between them.

*It would be helpful for the reader to explain the “Hogrefe” system

Discussion

*How are differences in symptom distress related to diagnosis?Is there an interaction effect of diagnosis and QoL?

*It may be helpful to also consider measurement error for current or previous research findings. Is there gender bias in the standardized tools?

*The statement “The present findings confirm that gender-sensitive treatment is possible, even if women are still more distressed than men upon admission.” Is not well supported by your research question or results. Rather, it seems to suggest that rehabilitation interventions are effective despite limited access to gender-sensitive treatment. It may also suggest looking at measurement tools differently to capture gender differences and/or the use of qualitative exploration to reveal gender differences.

* There is also evidence that people in the age cohorts of younger and older adulthood may require age-specific treatment or gender-sensitive treatment based upon age.

6. PLOS authors have the option to publish the peer review history of their article (what does this mean? ). If published, this will include your full peer review and any attached files.

Reviewer #1: No

Reviewer #2: No

---

## [Author Response · Author response to Decision Letter 0]

10 Aug 2021

Dear Dr. Doering, Dear Reviewers,

Thank you very much for allowing us to revise our submission and for the very helpful comments. We have carefully revised the manuscript accordingly.

Sincerely,

Juliane Burghardt

Reviewer #1: Summary and overall impression

The present study investigated whether gender differences in the effectiveness of treatment of mental illness are evident in psychosomatic rehabilitation. Previous studies showed such an effect, although the evidence for this has been less clear in studies conducted in recent years. The present work shows that gender differences are partly present but clinically negligible. Differences are evident in very young and very old patients, although also small. The authors discuss different reasons why gender differences are less apparent compared with the past. The paper thus represents an important work in the field of psychosomatic rehabilitation, as it demonstrates that treatment is almost as effective for women as for men compared to the past. The sample size in all three age cohorts and the inclusion of measures of self-assessment and external assessment in the analysis are strengths of the study. Nonetheless, the fact that there are many more individuals in the middle age cohort would be important to address more in the discussion. This uneven distribution of age cohorts has a large impact on the overall lack of differences in the sample. As the authors themselves describe, it would also have been desirable if statements about gender role orientation had been possible, as these appear to be highly relevant to the interpretation of the results. Overall, the paper is very clear and well structured. It provides interpretative information on many relevant points and represents an important gain in knowledge concerning the effectiveness of psychosomatic rehabilitation.

Specific improvements

Major issues

- The discussion should strongly consider that the middle age group contained significantly more individuals, which reduced the gender effects so much in the overall analysis, although there are at least small differences in the other two age groups. I would like to see a more nuanced discussion of this.

++ We have added this to the discussion:

“Since most patients in this study fall within the middle age group the very small overall gender effect is driven by this age group. However, previous meta-analyses of psychosomatic rehabilitation interventions show that this age distribution is representative of psychiatric rehabilitation clinics in Austria and psychosomatic rehabilitation clinics Germany [3, 33].” (p. 12)

Minor issues

- Page 3 “Among other things, seeking help […]“: n my view, the sentence does not fit well as an explanation for the effects of gender role in rehabilitation, since the individuals have already sought help and started rehabilitation. Or are there findings that men then take the treatment itself less seriously or make less use of it during inpatient treatment?

++ We agree and have clarified the explanation.

“Among other things, seeking help and support contradicts the male gender role, which is why men often show less help seeking behavior than women [10] and communicate less effectively with health care providers. For instance, some men may understate pain or hide emotions [8] or are less accurate about their medical records [11]. The loss of autonomy during treatment is possibly more problematic for men than for women [12]. The patient role is incongruent with the male gender role [13], which is active and agentic. In line with the male gender role, men prefer a higher level of control over the therapeutic process and report a need for action-oriented problem-solving strategies [12].” (p. 3-4)

- Page 4: “Women suffer more often […]“: This sentence seems in the wrong place, since it seems that it does not contribute to the linked arguments in the sentences before and after this sentence.

++ We have deleted the sentence.

- Page 6: I would recommend to put the G*Power analysis together with a more detailed description of the conducted analyses in the Analyses section. Especially the repeated-measure ANOVAs could be described more in detail.

++ We have restructured the method section and included the power analysis in the analysis section.

- Page 9: “The biggest effects were found in for functional ability […]”

++ We have corrected that.

- Page 10: “[…] men showed the better course in this measure, which might be because […]”: Since the second half of this sentence is an interpretation, it should be moved to the discussion.

++ We have removed the interpretation from the result section.

- Page 24: Table 2 needs some revision in the section “Employment”. The percentage of employed people in the oldest WHO group is in the wrong line and is missing it’s percent sign.

++ We have revised the Table.

- The authors should point out in the limitations, that the data was obtained only from one clinic and therefore generalization is limited. Furthermore, they should discuss that the data of some patients is ten years old and gathered over a long period of time, where e.g. changes in the understanding of gender roles in the society could have changed slightly and therefore confounded the analyses.

++ We have revised the limitation accordingly.

“Additionally, the data was collected between 2011 and 2015. It is thus, plausible that new changes have occurred. This and the fact that the data was only gathered in one clinic makes it necessary to replicate our findings.” (p. 13)

Reviewer #2: This is an interesting paper about the effects of gender on the treatment outcomes of inpatient psychiatric rehabilitation in Austria. The paper is fairly clear in the approach and analysis. The manuscript would benefit from a few changes for readability and clarity.

Introduction

*Please describe psychosomatic rehabilitation- how is this the same or different from psychiatric rehabilitation? It may be helpful to use either one term or the other- the term “psychiatric” is preferred to describe hospital treatment.

++ We have added explanations for the similarities and differences between psychosomatic and psychiatric rehabilitation interventions. In fact, what is called psychosomatic rehabilitation in Germany is very similar to what is called psychiatric rehabilitation in Austria. However, since these terminological differences are somewhat political in Austria, we would prefer to use both terms.

Both psychosomatic and psychiatric rehabilitation involve inpatient hospital treatments.

“Psychosomatic rehabilitation in Germany and psychiatric rehabilitation in Austria are very similar approaches to treat mental disorders. Both emphasize psychotherapeutic interventions, but also include psychopharmacological and various other complementary interventions, for instance excise and physical therapy. Both mainly treat patients with depressive or anxiety-related disorders.” (p. 8)

* Please use rehabilitation interventions not rehabilitations

++ Thanks, we did.

p.4 Consider mentioning the subjective nature diagnosis – are there studies that examine the effect of clinician gender on treatment outcomes?

++ In inpatient treatment it difficult to identify the clinician’s gender because patients are treated by multiple health care providers (i.e., psychotherapists, psychiatrics, physical therapist, art and music therapists). Previous findings suggested that patient therapeutic dyads with the same gender could be more effective, however, since inpatients are treated by multiple individuals it seems difficult to generalize these results to inpatients.

The diagnosis is merely included to help describe the sample.

Method

*There is some duplication in description of study design.

++ We revised the method section to be less repetitive.

*Please address why the BriefSCL and SCLwere both utilized and the comparability of results between them.

++ We added information regarding the comparability of the two questionnaires. It’s unclear why the measures were changed, maybe for length. The persons likely responsible for this decision are not available.

“The BSCL had been derived from the SCL-90 by selecting the items with the highest item loading. Previous data showed that the BSCL and SCL-90 are highly correlated [r = .92 to .99, 30]. Both instruments provide a Global Severity Index (GSI), which is a commonly used measure of general psychological distress.” (p. 7)

*It would be helpful for the reader to explain the “Hogrefe” system

++ We have added information about the Hogrefe system.

“The self-report measures were assessed using the Hogrefe test system. This system provides a platform that administers standardized questionnaires licensed by Hogrefe. It provides a user-friendly surface and assures data integrity.” (p. 7)

Discussion

*How are differences in symptom distress related to diagnosis? Is there an interaction effect of diagnosis and QoL?

++ That is a very important question. However, we believe that an analysis of the relations between symptom distress and diagnosis and QoL would be too complex to add to this manuscript. The diagnosis should just help to describe the sample.

*It may be helpful to also consider measurement error for current or previous research findings. Is there gender bias in the standardized tools?

++ We have added a discussion of a gender bias within the measurement tool.

“Previous work has noted that the gender differences could be partly attributed to biases within measures [18]. Since men and women often report different symptoms questionnaires can be biased towards capturing traditionally ‘feminine’ symptoms of a disorder. Therefore, it is notable that our study used the SCL-90 lists, which is the most common instrument used in studies within the MESTA-meta-analysis, which did find a gender effect β = .22 [3]. It is therefore unlikely that our results hinge on measurement biases.” (p. 12-13)

*The statement “The present findings confirm that gender-sensitive treatment is possible, even if women are still more distressed than men upon admission.” Is not well supported by your research question or results. Rather, it seems to suggest that rehabilitation interventions are effective despite limited access to gender-sensitive treatment. It may also suggest looking at measurement tools differently to capture gender differences and/or the use of qualitative exploration to reveal gender differences.

++ We agree, we have corrected the sentence to be more precise.

“The present findings confirm that rehabilitation interventions can be equally effective among men and women, even though women remain more distressed upon admission than men. Thus, the higher symptom burden of women at admission does not necessarily lead to a difference in treatment effectiveness. It remains unclear whether the decline in gender differences on treatment effectiveness was caused by changes in society as a whole or by specific characteristics of the present treatment.” (p. 14)

* There is also evidence that people in the age cohorts of younger and older adulthood may require age-specific treatment or gender-sensitive treatment based upon age.

++ We have added this to the discussion.

“If the finding of comparatively bigger gender effects among older or younger patients replicates, this might imply a need for gender and age-sensitive treatments instead of a gender-sensitive treatment that ignores age.” (p. 14)

---

## [Editor Report · Decision Letter 1]

19 Aug 2021

Gender effects on outcomes of psychosomatic rehabilitation are reduced

PONE-D-20-37367R1

Dear Dr. Burghardt,

We’re pleased to inform you that your manuscript has been judged scientifically suitable for publication and will be formally accepted for publication once it meets all outstanding technical requirements.

An invoice for payment will follow shortly after the formal acceptance. To ensure an efficient process, please log into Editorial Manager at http://www.editorialmanager.com/pone/ , click the 'Update My Information' link at the top of the page, and double check that your user information is up-to-date. If you have any billing related questions, please contact our Author Billing department directly at authorbilling@plos.org.

Kind regards,

Stephan Doering, M.D.

Academic Editor

PLOS ONE

---

## [Editor Report · Acceptance letter]

20 Aug 2021

PONE-D-20-37367R1

Gender effects on outcomes of psychosomatic rehabilitation are reduced

Dear Dr. Burghardt:

I'm pleased to inform you that your manuscript has been deemed suitable for publication in PLOS ONE. Congratulations! Your manuscript is now with our production department.

Kind regards,

on behalf of

Professor Stephan Doering

Academic Editor

PLOS ONE